# Characteristics of Soil Nutrients and Their Ecological Stoichiometry in Different Land Use Types in the Nianchu River Basin

**Yanling Liu** [1,2], **Lijiao Fu** [1,2], **Xuyang Lu** [1] **and Yan Yan** [1,*]

[1] Key Laboratory of Mountain Surface Processes and Ecological Regulation, Institute of Mountain Hazards and Environment, Chinese Academy of Sciences, Chengdu 610042, China; liuyanling@21.mails.ucas.ac.cn (Y.L.); fulijiao@imde.ac.cn (L.F.); xylu@imde.ac.cn (X.L.)

[2] University of Chinese Academy of Sciences, Beijing 101408, China

[*] Correspondence: yy@imde.ac.cn; Tel.: +86-189-8082-5169

**Abstract:** Land use types can cause changes in soil chemical characteristics, such as altering soil C, N, and P contents and distribution. The aim of this study was to investigate the status of soil C, N, P and other nutrient contents and their stoichiometric ratios in the terrestrial ecosystem of Nianchu River Basin, Tibet. A total of 102 sample plots with 306 soil samples and 102 plant samples were investigated in August 2021 along the Nianchu River basin by selecting four land-use types: grassland, shrubland, forestland, and farmland. The soil's basic physical and chemical properties (soil organic matter (SOM), total nitrogen (TN), total phosphorus (TP), alkaline nitrogen (AN), available phosphorus (AP), pH, and soil particle composition) were examined at each sampling point, and the stoichiometric characteristics of C, N, and P of the soils were analyzed using one-way analysis of variance (ANOVAs). The results revealed that the C and N contents of shrubland were significantly lower than those of grassland, forestland, and farmland, with farmland having the highest P content. For all land types, C:N increased with increasing soil depth, while C:P and N:P decreased with increasing soil depth. PCA and RDA analyses revealed that soil texture and pH had an impact on soil C, N, and P contents, as well as stoichiometric ratios.

**Keywords:** Qinghai–Tibetan Plateau; various land-use types; Soil C, N, P content; ecological stoichiometry; C:N:P ratio

## 1. Introduction

Carbon (C), nitrogen (N), and phosphorus (P) are the three primary elements that drive nutrient cycling and plant growth in ecosystems [1–3]. The balance of nutrients is known as ecological stoichiometry [4], and is critical in the study of primary productivity in terrestrial ecosystems [5]. Changes in soil C, N, and P contents and soil properties can alter C:N:P stoichiometry, affecting the structure and function of ecosystems [1]. Ecological stoichiometry provides a conceptual framework for terrestrial ecological studies [6], and current research has focused on the effects of ecological stoichiometry on plant growth and ecosystems, with little understanding of soil stoichiometry and nutrients [7]. As a result, studying soil C, N, and P contents and stoichiometry in various ecosystems has become a critical task. In ecosystems, the C:N:P ratio influences primary production, nutrient cycling, and food web dynamics [2]. C:N and C:P ratios can reflect organic matter decomposition rates, nutrient mineralization or immobilization, and plant nutrient limitations [8,9]. N:P is an effective nutrient limitation indicator, and N and P are generally the limiting elements for plant growth [5]. Human activities (e.g., fertilizer application) and climatic factors (e.g., temperature and precipitation) influence soil C:N:P [10].

Land-use types affect soil physicochemical properties [11,12] and the biogeochemical cycling of C, N, and P [13] via debris and apoplastic decomposition, and nutrient inputs and

outputs [14]. Soil nutrients differ significantly across land-use types. Tian et al., 2009, and Liu and Wang., 2020, for example, investigated alpine regions such as the Qinghai–Tibet Plateau and the Loess Plateau and discovered that C and N contents, C:P, and N:P were higher in forestland than in grassland and farmland, and pH was lower in forestland than in grassland and farmland [15,16]. Liu et al., 2017, found that P content, C:N, C:P, and N:P were lower in grassland soils of the Yili River Valley in Xinjiang than in shrubland soils [17]. Wang et al., 2018, observed that the C, N, and P contents of grassland and cropland in the eastern Tibetan Plateau were higher than those of forestland [18].

The Nianchu River basin is a significant part of Tibet's "One River, Two Streams" region, it is the largest among the five major tributaries of the Yarlung Tsangpo River in Tibet, and its valley is a developed agricultural area with unique geographical and climatic conditions. In the background of global warming, the basin and other high-altitude areas are environmentally fragile and sensitive, and the ecological environment is deteriorating [19]. The Nianchu River basin is facing a series of ecological and environmental problems such as soil erosion, degradation of grassland and arable land, overgrazing of pastures, and destruction of forest trees [19]. The basin's ecological environment is in desperate need of improvement. This study is critical for acquiring a thorough understanding of the basin's current ecological environment, subsequent ecosystem restoration and protection, and providing scientific data for the protection of the Tibetan Plateau ecosystem. A number of studies have been conducted on the Nianchu River basin in recent years, primarily on the community characteristics of wetland soil fauna in the basin [20], hydrological characteristics of the basin and its response to climate change [21], and changes in runoff and glaciers and their influencing factors [22–24]. However, there have been fewer studies on terrestrial ecosystems in the basin, particularly soils. Therefore, four land-use types, namely, forestland, shrubland, grassland, and farmland, were chosen in this paper. We aimed to (1) study soil C, N, and P contents and their stoichiometry of the four land-use types in the Nianchu River basin, (2) assess the mechanism of the impact of land use type on soil nutrients and their stoichiometry, and (3) summarize the implications of soil nutrient changes in the basin for further ecological restoration.

## 2. Materials and Methods

### 2.1. Study Area

The Nianchu River basin, located at 88°35′ E–90°15′ E and 28°10′ N–29°20′ N, involving Kangma County, Gyantse County, Bailang County, and the Sangchuze District of Rikaze City, Tibet, is a first-class tributary of the five major tributaries of the Yarlung Tsangpo River, with a 11,130 km$^2$ basin area, and 217 km river length. The basin has a plateau, temperate, semi-arid monsoon climate with an annual precipitation of about 365 mm and an annual temperature of 4–6 °C. *Hordeum vulgare*, *Triticum aestivum*, *Brassica napus*, and other crops were primarily planted in the basin. There are also forestlands and shrubland in the valley, with the dominant species being *Salix xizangensis*, *Hippophae tibetana*, and *Hippophae gyantsensis*. The dominant species in the alpine meadow grassland are *Kobresia pygmaea* and *stipa purpurea*. According to the Chinese soil classification standards, the basin soils are semi-Luvisols, dark semi-hydromorphic soils, and hydromorphic soils, etc.

### 2.2. Soil Sampling and Data Sources

In August 2021, 34 sampling areas (100 × 100 m) (3 forestlands, 18 grasslands, 10 farmlands, and 3 shrublands), were selected using a fixed grid for the survey sample plots, dividing the entire Nianchu River basin into grids according to 20 × 20 km, and selecting typical terrestrial ecosystem types. With three replicate sample squares (10 × 10 m for forestland, 5 × 5 m for shrubland, and 1 × 1 m for grassland and farmland), 102 sampling points were established along the Nianchu River valley alignment (Figure 1). After surveying each sampling site for above-ground plant species, above-ground plant leaves were cut, placed in envelopes, and returned to the laboratory to be washed with distilled water, air-dried, and dried in an oven at 60 °C, then weighed using an analytical

balance. We then determined their C, N, and P contents after grinding finely to 0.15 mm with a ball mill [25].

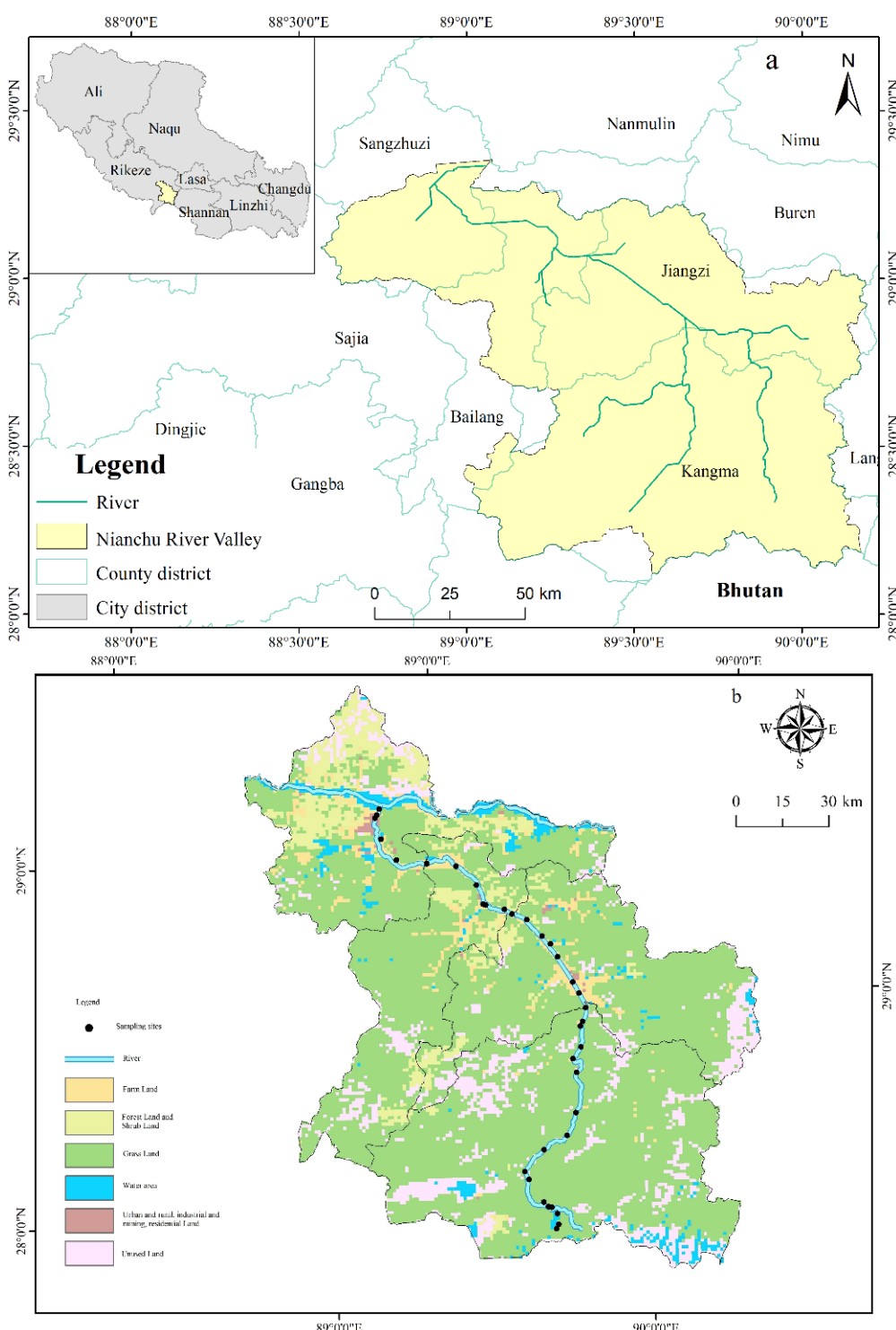

**Figure 1.** Diagram of sampling sites in the Nianchu River basin. (**a**) Geographic location map, (**b**) Sampling site map.

For a total of 306 soil samples, soil from 0 to 30 cm was collected in three layers of 0–10 cm, 10–20 cm, and 20–30 cm and placed in self-sealing bags. To ensure soil purity, surface debris such as apoplast was removed before sampling, and the five-point sampling method was used. Five soil cores were mixed at each depth to create a composite soil

sample, and three soil samples (0–10 cm × 1, 10–20 cm × 1, and 20–30 cm × 1) were obtained from each sample. The soil types analyzed were meadow soils, Fluvo-aquic soils, Grey-cinnamon soils, and Bog soils. The soil samples were returned to the laboratory for natural air-drying and passed through a 2 mm sieve before physical and chemical analysis. The soil organic matter (SOM), total nitrogen (TN), total phosphorus (TP), alkaline nitrogen (AN), available phosphorus (AP), soil particle composition, and pH of the soil were all determined.

Soil pH was measured using a pH meter in a 1:2.5 soil–water dilution sample. Soil particle composition (clay, silt, fine sand, and sand) was measured by a laser particle-size analyzer, and soil organic matter (SOM) and plant C content was measured using a potassium dichromate oxidation method [26]. Soil and plant total phosphorus (TP) content was measured with the colorimetric method using ammonium molybdate after acid digestion. Soil available phosphorus (AP) content was measured with the colorimetric method using molybdate. Alkaline nitrogen (AN) was measured by the alkali diffusion method, and soil and plant total nitrogen (TN) was determined by the Kjeldahl method [11].

*2.3. Data Analysis*

Data were counted and processed in SPSS version 24.0 (SPSS, Inc., Chicago, IL, USA) and Excel version 2016(Microsoft, Washington, DC, USA). The mean, standard deviation (SD), and coefficient of variation (cV) were calculated to describe the basic characteristics of topsoil nutrients, and the data was also checked for normal distribution with K-S, which analyzed the raw data without any transformation. One-way ANOVAs (normally distributed data) and Mann-Whitney tests (non-normally distributed data) were performed on the data using SPSS version 24.0 (SPSS, Inc., Chicago, IL, USA) software for analyzing the effect of land use type on soil nutrients and their chemometric characteristics. The effects of land use type and soil depth on soil physicochemical properties and stoichiometric characteristics were determined using two-way ANOVAs. Origin version 2021 (OriginLab, Northampton, MA, USA) was used to conduct a correlation analysis to look into the relationship between soil properties, plant nutrients, and soil and plant stoichiometric ratios. Origin version 2021 (OriginLab, USA) was used to perform principal component analysis (PCA) and using Canoco 5.0, RDA was performed and images were plotted to determine the relationship between basic soil physicochemical and soil stoichiometric ratios.

## 3. Results

*3.1. Changes in Soil SOM, N and P Contents under Different Land Use Types*

The basin SOM, TN, AN, TP, and AP contents were 8.56–24.40 g·kg$^{-1}$, 0.40–1.26 g·kg$^{-1}$, 9.31–36.21 mg·kg$^{-1}$, 0.49–0.62 g·kg$^{-1}$, and 1.49–11.83 mg·kg$^{-1}$, respectively, with coefficients of variation (Cv) of 30.68–54.36%, 30.02–61.06%, 34.20–78.26%, 4.73–22.51%, and 40.54–89.51%, respectively. The basin soil pH was 7.95–8.38 and is slightly alkaline. As shown in Table 1, the SOM content in the 0–30 cm soil layer was significantly lower in the shrubland than in the grassland, forestland, and farmland. The TN content of shrubland was significantly lower than that of grassland, forestland, and farmland, and that in forestland was significantly lower than that in farmland. The AN content of shrubland was significantly lower than that of grassland, forestland, and farmland, and significantly higher in grassland and farmland than in forestland. Farmland and forestland had a significantly higher TP content than shrubland and grassland. The AP content of farmland was significantly higher than that of shrubland, forestland, and grassland, and shrubland content was significantly lower than that of grassland and forestland. The pH of shrubland was significantly higher than that of grassland, forestland, and farmland. Additionally, the pH of grassland significantly higher than that of farmland.

**Table 1.** Characteristics of C, N, and P contents and their stoichiometric ratios in 0–30 cm soil layers of different land use types.

| Indicators | Grassland | | | Shrubland | | | Forestland | | | Farmland | | |
|---|---|---|---|---|---|---|---|---|---|---|---|---|
| | SD | Av | Cv (%) | SD | Av | Cv (%) | SD | Av | Cv (%) | SD | Av | Cv (%) |
| pH | 0.20 | 8.15 A | 2.50 | 0.21 | 8.28 B | 2.55 | 0.18 | 8.05 AC | 2.22 | 0.22 | 8.05 C | 2.74 |
| SOM (g·kg⁻¹) | 10.78 | 20.93 A | 51.52 | 6.02 | 11.08 B | 54.36 | 10.03 | 18.80 AC | 53.37 | 5.42 | 17.68 AC | 30.68 |
| TN (g·kg⁻¹) | 0.61 | 1.10 AC | 55.18 | 0.29 | 0.53 B | 55.65 | 0.54 | 0.89 A | 61.06 | 0.30 | 1.01 C | 30.02 |
| TP (g·kg⁻¹) | 0.11 | 0.50 A | 22.51 | 0.02 | 0.52 A | 4.73 | 0.04 | 0.57 B | 6.98 | 0.09 | 0.60 B | 15.73 |
| AN (mg·kg⁻¹) | 18.80 | 30.64 A | 61.37 | 10.27 | 13.12 B | 78.26 | 16.05 | 22.28 C | 72.06 | 9.16 | 26.78 AD | 34.20 |
| AP (mg·kg⁻¹) | 1.05 | 2.58 A | 40.54 | 0.94 | 1.96 B | 47.72 | 1.43 | 2.84 A | 50.37 | 7.28 | 8.13 C | 89.51 |
| C:N | 2.33 | 11.50 A | 20.23 | 1.02 | 12.36 B | 8.24 | 1.20 | 12.50 B | 9.59 | 0.96 | 10.17 C | 9.42 |
| C:P | 12.47 | 25.18 A | 49.55 | 6.51 | 12.38 B | 52.58 | 9.64 | 19.07 C | 50.56 | 5.61 | 17.29 C | 32.43 |
| N:P | 1.18 | 2.24 A | 52.65 | 0.54 | 1.01 B | 53.70 | 0.88 | 1.55 C | 56.73 | 0.50 | 1.45 C | 34.31 |
| AN:AP | 8.20 | 13.20 A | 62.11 | 4.67 | 7.54 BC | 61.98 | 4.37 | 8.36 B | 52.23 | 3.84 | 5.34 C | 71.87 |

Note: Different capital letters indicate significant differences among different land types in 0–30 cm soil layer ($p < 0.05$) SOM: soil organic matter; TN: total nitrogen; TP: total phosphorus; AN: alkaline nitrogen; AP: available phosphorus; C: N: carbon to nitrogen ratio; C:P: carbon to phosphorus ratio; N:P: nitrogen to phosphorus ratio; AN:AP: alkaline nitrogen to available phosphorus ratio; SD: standard deviation; Cv: coefficient of variation; and Av: Average value. Same as below.

Figure 2 shows that there was no statistically significant difference in the pH, SOM, and N content between land-use types (Figure 2a–d). The TP content in the 0–10 cm soil layer was significantly higher in farmland and forestland than in grassland and shrubland (Figure 2e). Farmland had a significantly higher AP content than grassland and shrubland (Figure 2f). Shrubland had significantly lower SOM and TN contents in the 10–20 cm soil layer than grassland and farmland (Figure 2b,c). Farmland and grassland had a significantly higher AN content than forestland and shrubland, and forestland had a significantly higher AN content than shrubland (Figure 2d). Farmland had a significantly higher TP content than grassland and shrubland (Figure 2e). Farmland had a significantly higher AP content than forestland, shrubland, and grassland, and grassland had a significantly higher AP content than shrubland (Figure 2f). Shrubland had significantly lower SOM, TN, and AN content in the 20–30 cm soil layer than grassland and farmland (Figure 2b–d). Farmland had a significantly higher AN content than forestland (Figure 2d). The TP content was significantly higher in farmland than in grassland and significantly higher in forestland than in shrubland, and the AP content was significantly higher in farmland than in grassland, shrubland, and forestland (Figure 2e). The pH shrubland was significantly higher than grassland, forestland, and farmland in the 10–20 cm and 20–30 cm soil layers (Figure 2a).

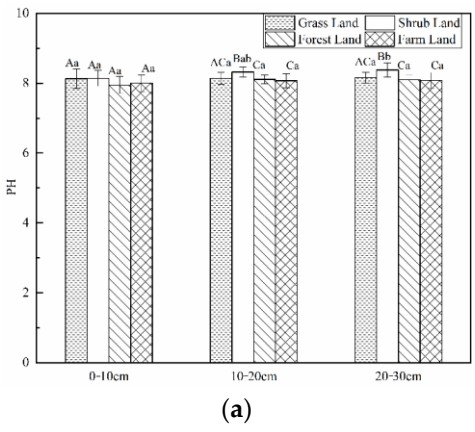

(a)

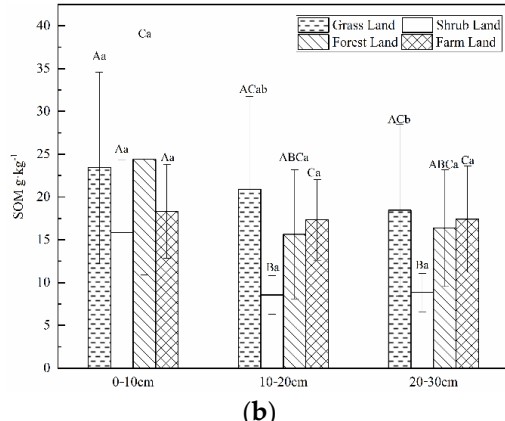

(b)

**Figure 2.** *Cont.*

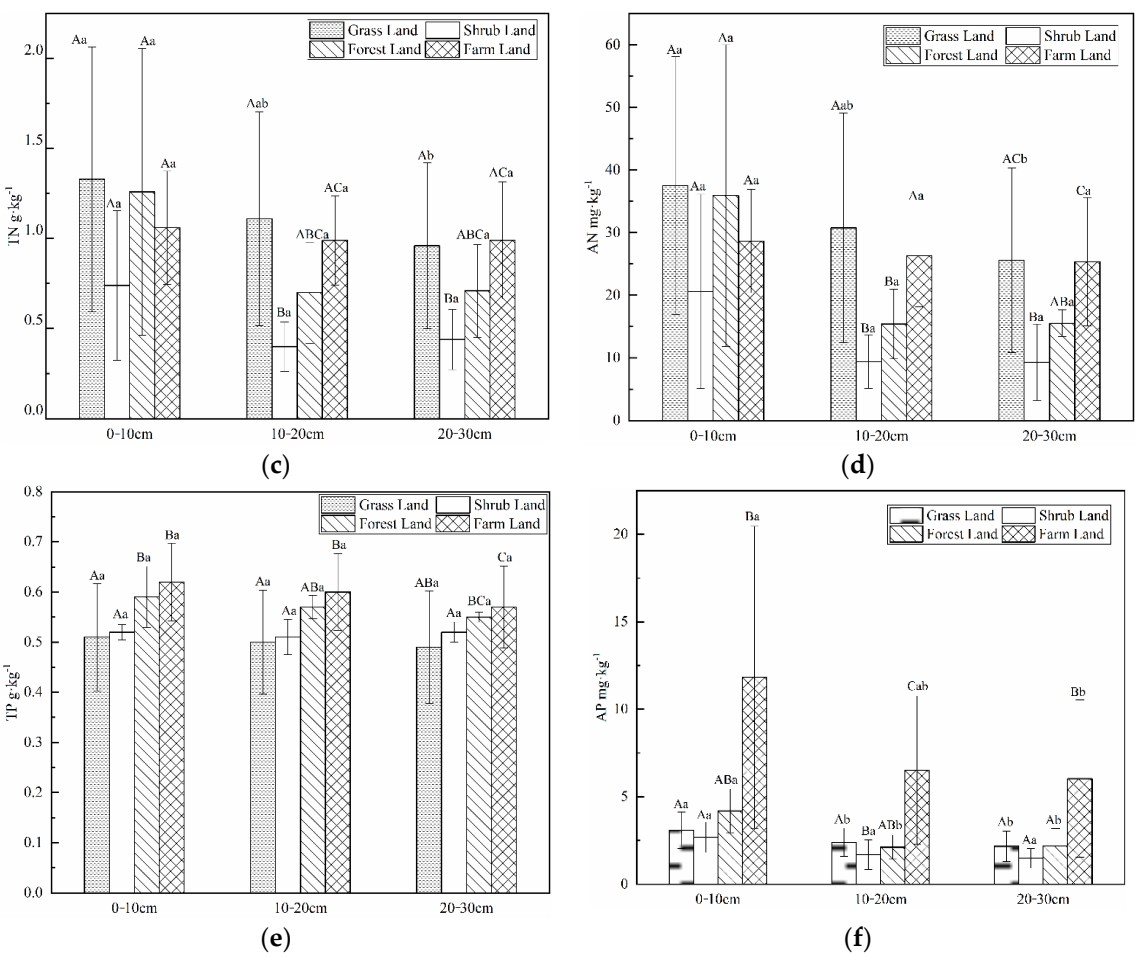

**Figure 2.** Differences in SOM, TN, TP, AN, and AP contents and pH values in grassland, shrubland, forestland, and farmland. (**a**) pH value, (**b**) SOM content, (**c**) TN content, (**d**) AN content, (**e**) TP content, (**f**) AP content. Error bars indicate the standard error of the mean. Different capital letters indicate significant differences between land use types ($p < 0.05$); different lowercase letters indicate significant differences between soil depths ($p < 0.05$).

### 3.2. Analysis of Soil Nutrient Stoichiometric Ratios under Different Land-Use Types

The basin's soil C:N, C:P, and N:P ranges were 10.07–13.31, 9.78–27.10, and 0.79–2.46, respectively. Variation coefficients (Cv) were 8.24–20.23%, 32.43–52.58 %, and 34.31–56.73%, respectively. The type of land use had a significant impact on soil C:N, C:P, and N:P ratios (Table 2). As shown in Table 1, soil C:N in the 0–30 cm soil layer was significantly lower in farmland than in grassland, shrubland, and forestland, and significantly higher in forestland and shrubland than in grassland. C:P and N:P levels in grassland were significantly higher than in shrubland, farmland, and forestland, and significantly higher in forestland and farmland than in shrubland. The ratio of AN:AP in grassland was significantly higher than in shrubland, forestland, and farmland, and significantly higher in forestland than in farmland.

**Table 2.** Analysis of variance (ANOVA) of soil nutrients and their stoichiometric ratios for different land use types.

|  | PH | TN | TP | SOM | AN | AP | C:N | C:P | N:P | Clay | Silt | Fine Sand | Sand |
|---|---|---|---|---|---|---|---|---|---|---|---|---|---|
| **Land use type** | *** | *** | *** | *** | *** | *** | * | *** | *** | *** | *** | *** | NS |
| **Depth** | NS | * | * | NS | ** | *** | NS | NS | *** | NS | NS | NS | NS |

\* $p < 0.05$, \*\* $p < 0.01$, and \*\*\* $p < 0.001$; NS, Nonsignificant.

As shown in Figure 3, the soil C:N in the 0–10 cm soil layer was significantly lower in farmland than shrubland and grassland. The soil C:P in grassland was significantly higher than that in shrubland, farmland, and forestland, and significantly higher in forestland than in farmland and shrubland. The soil N:P and AN:AP in grassland were significantly higher than those in farmland. In the 10–20 cm soil layer, the C:N in farmland was significantly lower than that in grassland, shrubland, and forestland. The C:P and N:P were significantly lower in shrubland than in farmland and grassland, and the C:P in farmland was significantly lower than that in grassland. The AN:AP in grassland was significantly higher than farmland. In the 20–30 cm soil layer, the C:N was significantly lower in farmland than in grassland, shrubland, and forestland, and significantly higher in forestland than in grassland. The C:P was significantly lower in shrubland than in grassland, forestland, and farmland. The N:P was significantly higher in grassland than in shrubland and farmland. The AN:AP was significantly higher in grassland than in farmland. As seen in Table 2, the soil C:N increased with increasing soil depth, and the soil C:P and N:P decreased with increasing soil depth.

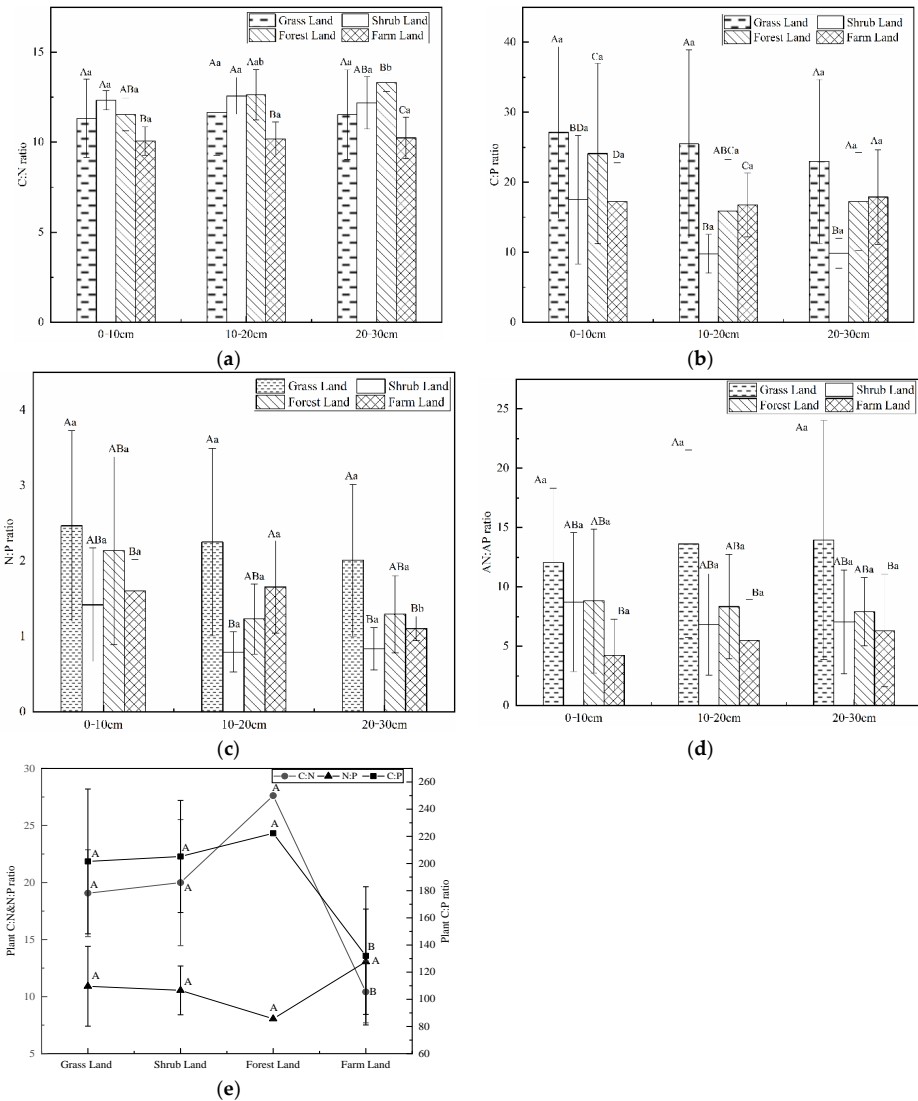

**Figure 3.** Differences in soils and their phytochemical stoichiometry in grassland, shrubland, forestland, and farmland. (**a**) Soil C:N ratio, (**b**) Soil C:P ratio, (**c**) Soil N:P ratio, (**d**) Soil AN:AP ratio, (**e**) Plant C:N, C:P and N:P ratio. Different capital letters indicate significant differences between land use types ($p < 0.05$); different lowercase letters indicate significant differences between soil depths ($p < 0.05$).

### 3.3. Relationship between Soil Nutrient Stoichiometric Ratios and Basic Physicochemical Properties

Figure 4 depicts the PCA results for the basic soil physicochemical properties and their stoichiometric ratios. The PCA1 and PCA2 axes explained 48.3% of the total variance. Fine sand, sand content, and pH could explain the variability in the C:N:P stoichiometric ratios (Figure 4). The soil properties accounted for 76.21% of the total variance in the C, N, and P contents, and stoichiometric ratios derived from redundancy analysis (RDA) (Figure 5). The soil C, N, and P contents and C:P, N:P, and pH were highly correlated with the first axis, while the soil C:N, clay, fine sand, and silt contents were highly correlated with the second axis. Figures 4 and 5 show that the fine sand and soil C:N are in the same direction and have a positive effect; clay and silt are in the opposite direction and have a negative effect. The pH and soil TN, TP, AN, AP, SOM, C:P, and N:P are in the opposite direction and are negatively correlated.

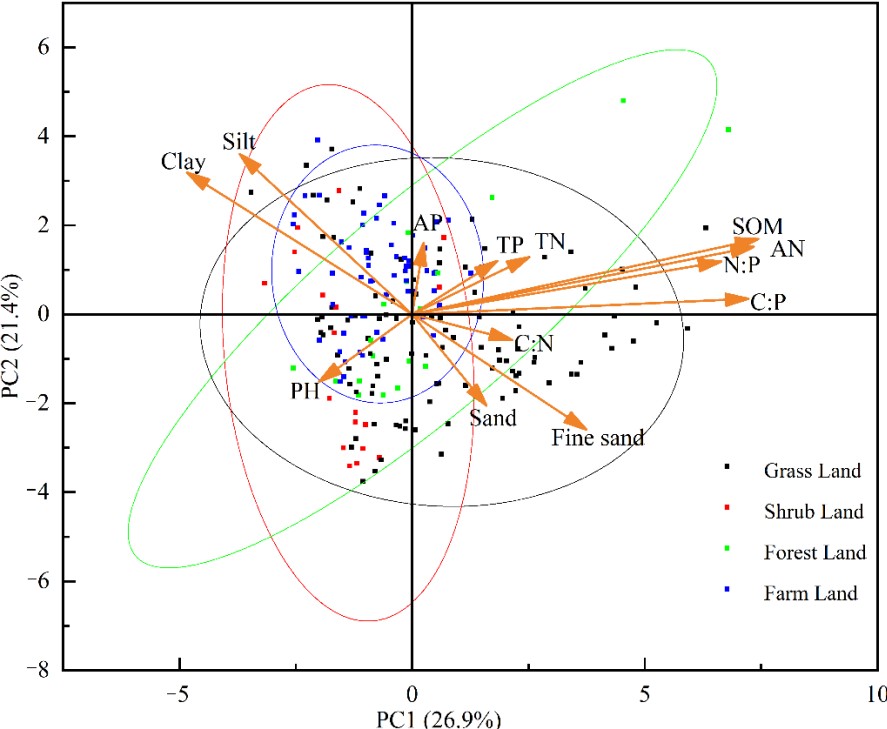

**Figure 4.** Double-labeled plots of the first two components of principal component analysis (PCA) under different land uses. Black, red, green, and blue circles indicate 95% confidence levels for grassland, shrubland, forestland, and farmland.

The basic physical and chemical properties of the soil and plant C, N, and P contents were significantly correlated with the stoichiometric ratios (Figure 6). The SOM was significantly positively correlated with the AN, soil C:N and N:P, and plant TN and TP, and negatively correlated with the pH and plant C:N and C:P. The TN was found to be significantly positively correlated with the TP and soil N:P. The AN was significantly positively correlated with the soil C:P and N:P, and plant TP, and significantly negatively correlated with the plant C:P. The AP was significantly positively correlated with the C:N and significantly negatively correlated with fine sand, and the plant TN, OC, and N:P. The plant TN was found to be significantly positively correlated with the soil C:P and N:P, fine sand, and plant TP, OC, and N:P, and significantly negatively correlated with the plant C:N and C:P. The plant TP was significantly positively correlated with the soil C:P and N:P and plant OC, and negatively correlated with the plant C:P and N:P. The plant OC was also significantly positively correlated with the soil C:P. The soil C:P and N:P were significantly positively correlated with fine sand. The soil C:P was significantly positively

correlated with the N:P and significantly negatively correlated with clay and the plant C:P and C:N. The N:P was significantly negatively correlated with the soil pH and plant C:P and C:N. The plant C:N was significantly positively correlated with the plant C:P. Fine sand was significantly negatively correlated with the plant C:N and C:P. The plant C:P was significantly positively correlated with the soil pH. The plant N:P was significantly negatively correlated with the plant C:N.

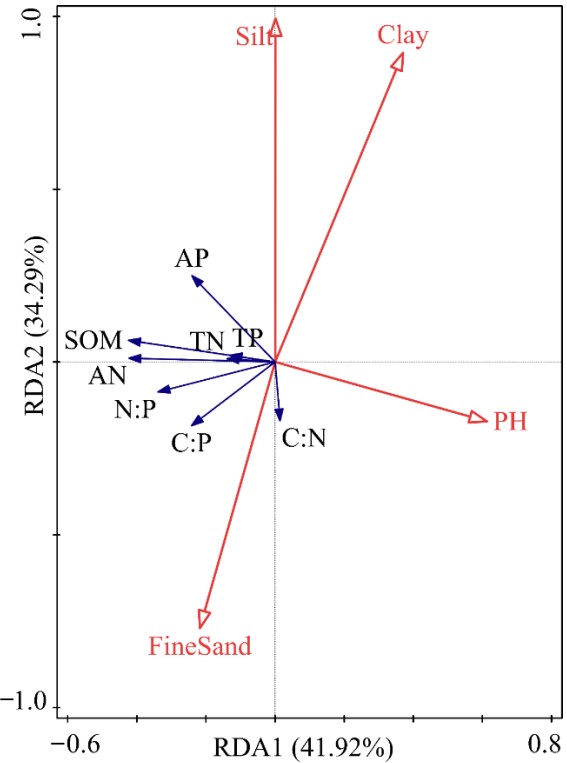

**Figure 5.** Two-dimensional redundancy analysis (RDA) sequence plots between soil properties and carbon, nitrogen, and phosphorus stoichiometric ratios for different land use types. Silt: 39.3%, F = 13.4 ; pH: 33.9%, F = 10.5 ; and Clay: 24%, F = 7.7.

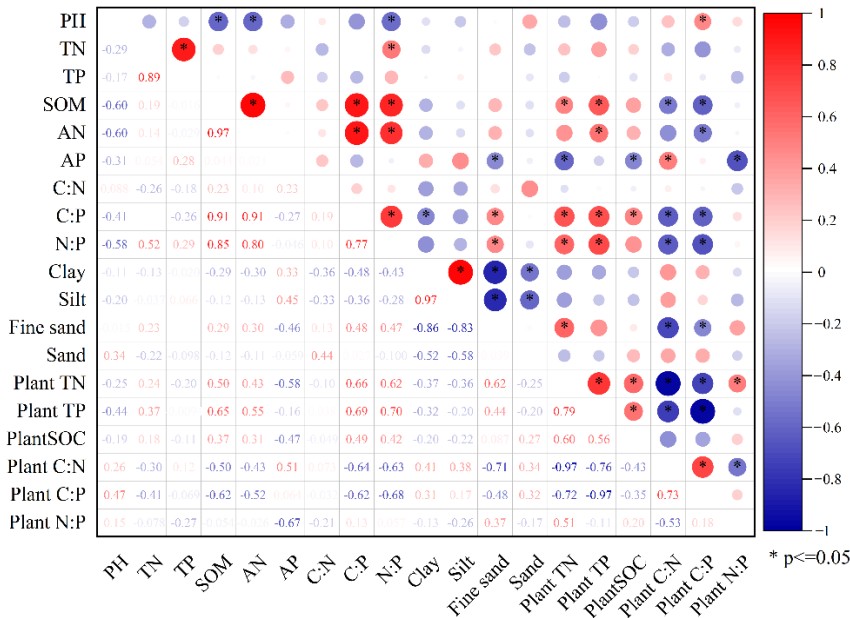

**Figure 6.** Correlation diagram of C, N, and P contents and their stoichiometric ratios.

## 4. Discussion

### 4.1. Effect of Land-Use Type on Soil Nutrients

The results of this study found that land-use type significantly influenced soil physicochemical properties [27–29] and stoichiometric ratios. Soil nutrients showed moderate variability (CV in the range of 10–100%) and basin soil nutrients showed strong spatial variability, according to the results of this study's coefficient of variation (Table 1). In this study, grassland SOM content was higher than forestland and farmland (Table 1), in contrast to Khan and Chiti (2022), who found higher SOM content in forestland than grassland in the Lithuanian region, and in agreement with Wang et al. (2018) and Zhang et al. (2013), who found higher SOM content in grasslands than in farmlands. This could be because grassland within this basin is primarily distributed in Kangma County, where the grassland type is primarily alpine meadow, with a thicker distribution of the grass felt layer and increased input of surface apoplankton, which provides more nutrients to the soil through decomposition [30]. At the same time, this type of meadow is rich in roots, and Wei et al., 2009, [31] discovered that meadows have more fine roots than forestland and shrubland in the 0–40 cm soil layer. Fine roots significantly contribute to soil organic matter [3], and their continuous production and decomposition [12,32] increase the SOC, resulting in the highest SOM content in the basin being found in grasslands. The SOC content of forestland is influenced by altitude, which is usually higher than grassland SOC content in areas below 2000 m in altitude [18]. The total elevation of the basin is greater than 3800 m. Higher elevations have lower net primary productivity and more water, so soil C input is lower [18]. Tillage disrupts soil structure, reduces the physical protection of soil organic carbon and N by soil structure, and accelerates the loss of original and newly imported organic carbon and N from the soil through a positive initiation effect [33], resulting in a large loss of organic carbon from farmland. However, the SOM and N contents in the 0–10 cm soil layer were higher in forestlands than in grasslands (Figure 2b–d), which differed from the findings of Wang et al., 2018 [18] in Qinghai Province. One of the reasons could be that the soil samples taken by Wang et al., 2018, were taken at a depth of 0–20 cm. The main reason, however, is due to the complicated mechanism of soil nutrient variation. The high SOM content in the forestland is due to the forestland apoplastic biomass is high at the soil surface, and the C and N content in the apoplastic layer is higher [34]. The shrubland had the lowest amount of SOM (Table 1), contradict the results of Zhang et al., 2013 [32]. The soil pH was found to be significantly negatively correlated with SOM in this study (Figure 6), because alkaline soils decompose soil organic matter faster than acidic soils [35], and pH can influence microbial activity [36] and bacterial community composition [37], thereby influencing organic matter turnover. The shrubland pH was the highest in this study, most likely due to the shrubland's low vegetation cover as a result of the basin's extensive deforestation for fuelwood [19] and low root biomass and residues, resulting in less decomposable organic matter, weaker microbial activity, and less carbon and organic acids. SOM and N distribution patterns are consistent across the profile [38], and the mechanisms underlying SOC and N loss are similar [39]. However, the N content of farmland did not conform to this pattern, and the N content of farmland was higher than that of forestland (Table 1), most likely due to the use of livestock manure and chemical fertilizers by local farmers to add nutrients to the farmland. Su et al., 2022, [33] demonstrated that the application of chemical and organic fertilizers increased soil N content and promoted plant growth, resulting in an increase in root debris returned to the soil and an increase in root biomass in the soil. Furthermore, soil SOM, TN, and AN content decreased by 36%, 45%, and 57%, respectively, in the 10–20 cm soil layer and 5%, 7%, and 8%, respectively, in farmland compared to the 0–10 cm soil layer in the forest. It can be seen that, as soil depth increases, the reduction in C and N content in farmland is lower than that in forest land, whereas N loss occurs more frequently in forestland. SOM and N content in the 0–10 cm soil layer did not differ significantly between land use types (Figure 2b–d). SOM and N in basin soils were between the third (SOM: 20–30 g/kg, N: 1–1.5 g/kg) and fourth (SOM: 10–20 g/kg, N: 0.75–1 g/kg) levels, and SOM and N were generally inadequate, according

to the nutrient grading requirements of the second national soil census. Surface soils have significantly different SOC and N levels than underlying soils, and this difference can be attributable to the effects of plant cover, apoplastic and root populations, and anthropogenic disturbances [32].

The TP content in forestland and farmland was higher than grassland and shrubland. In the Nianchu River Basin, P content was related to soil [26], and it can be shown that forestland and farmland soil P content was higher than grassland and shrubland. Grasslands had the lowest TP content, as discovered by Li et al., 2016, [34]. Basin shrublands often grow on grasslands, and shrublands mobilize more soil P than grasslands, increasing phosphorus depletion in the root profile and gaining more P from the soil, aggravating soil P depletion in grasslands [40]. Soil pH was lower and phosphatase activity was higher under grassland and secreted more acidic compounds and directly or indirectly (e.g., through feeding microbes) produced more phosphatase enzyme in the rhizo-sphere [40]. These mechanisms helped grassland plants acquire P from complex inorganic and organic P-containing compounds at the interface between the roots and soil [40].The AP content of grassland was higher than that of shrubland, implying that grassland had a higher P-use efficiency than shrubland because the phosphorus efficient-use strategy ensured that plants maintained photosynthesis and growth under low-phosphorus conditions [41]. The AP content of grassland was lower than that of forestland in the 0–30 cm soil layer, due to the positive correlation between AP and TP (Figure 6), and AP was influenced by soil TP [42]. However, the AP content of grassland was higher than that of forestland in the 10–20 cm and 20–30 cm soil layers, most likely due to the large, deep, root-induced gaps beneath the forestland, which could promote downward water movement and lead to increased phosphorus leaching [40]. Farmland had the highest P content due to the effect of phosphorus fertilizers and other phosphorus-containing substances that were applied during planting (e.g., manure and crop straw) [43]. Some fertilization experiments revealed that the addition of manure significantly increased the P content [38].

Under all land-use types, the topsoil layer (0–10 cm) had the highest levels of SOM, N, and P. As nutrients such as SOM and N generally accumulate in the soil's surface layer [30], microbially driven apoplast decomposition primarily occurs in the surface soil [14], increasing nutrient concentrations in the surface soil. Organic matter input decreases with increases in soil depth due to microbial decomposition activity and reduced root secretion [44]. In this study, there was no significant change in TP content with increasing soil depth for the four land-use types. Soil parent material, soil formation, tillage, and fertilization had the greatest influence on P content, and P migration in the soil was minimal [45]. The significant decrease in farmland AP with increasing soil depth could be attributed to the fact that farmland P is mostly inorganic P, which is easily soluble in water, resulting in a lower soil pH and higher phosphatase activity [40], both of which contribute to P turnover in the soil.

*4.2. Characteristics of Soil C, N, and P Stoichiometric Ratios in Different Land-Use Types*

The soil nutrient stoichiometry ratio is a critical indicator for assessments of soil nutrient supply capacity, soil quality, and function. The C:N of different land-use types ranged from 10.07 to 13.31 in this study, which is close to the average in China (11.9) [46], and the C:N and decomposition rate of soil organic matter were inversely proportional [47]. This could be because plant residues with a low C:N ratio, where mineralization and humification are easier, decompose faster, producing less humus and releasing more N elements. Soil C:N ratios were significantly lower in farmland than in grassland, shrubland, and forestland, supporting the findings of Yan et al., 2021 [48] and Huang et al., 2019 [49]. This is primarily due to the greater anthropogenic disturbance of farmland; under the long-term selection effect of human activities, fertilizer application is primarily chemical fertilizer with less organic fertilizer, crops are harvested above-ground after maturity, and the land is not tilled after harvesting, but livestock are driven into the area after harvesting crops for feeding for 1–2 months, which reduces the exogenous input of organic carbon to

cultivated land. This is consistent with Ma and Ping's research in the Nianchu River Basin, Tibet [50]. The C:N ratios were higher in forestlands and shrublands than in grasslands, supporting the findings of Gao et al., 2021 [40] and Li et al., 2016 [34]. This could be because persistent woody materials decompose SOM in soil at a slower rate than unstable N-containing compounds, and soil N losses under woody plants are higher than those under grasses, increasing C:N in forestlands. Meanwhile, forestlands are widely distributed in this basin, dominated by *Salix xizangensis*, *Hippophae tibetana*, and other trees and shrubs, and the more stable stand structure accumulated a rich, dead layer in the understory, which consists of materials that are more difficult to biochemically degrade (especially aliphatic biopolymers) and less suitable for use as microbial substrates than the dead material of residual grasses. This more recalcitrant apoplastic material may decompose more slowly, leading to a higher soil carbon-to-nitrogen ratio [51].

The soil C:P is an important indicator of soil phosphorus mineralization capacity, whereas N:P indicates soil N and P availability. In this study, soil C:P variation ranged from 12.38 to 25.18 and N:P variation ranged from 1.01 to 2.24 for basin land-use types, both of which are lower than the average values in China (C:P: 61.0, N:P: 5.2) [46]. Low C:P ratios increase soil C loss [48]. Plant growth is N-limited when N:P < 14 and soil C:N < 30 indicate a high risk of nitrate leaching [52]. This suggests that there may be some N loss and deficiency in the soils of terrestrial ecosystems in this basin. He et al., 2019 [53] discovered that N limitation was more common in cold biomes. Zhang et al., 2022 [54] discovered low C:P in farmland in Jiangxi Province (C:P: 7.18–132.19) and very low soil C:P in the Nianchu River basin farming environment. As a result, it is advised that basin soils be fertilized with greater nitrogen and less phosphorus due to straw return, combined application of chemical and organic fertilizers, and low SOC and C:P [54]. Soil C:N, C:P, and N:P were all considerably and positively connected with SOM and AN, and soil C:N was also significantly and strongly correlated with AP, according to correlation analysis (Figure 6). The findings of Zhang et al., 2022, are in line with ours [54], with grassland having the highest SOM and N content and the lowest TP content, resulting in the highest C:P and N:P, while high C:P resulted in less C loss in grassland than in forestland, shrubland, and farmland. Basin shrubland had a significantly lower SOM and N than forestland and farmland, and soil C:P and N:P were significantly positively correlated with soil SOM and AN (Figure 6). Faster-growing plants had higher N and P concentrations according to Zechmeister-Boltenstern et al., 2015 [6]. C:N and C:P were significantly higher in shrubland plants than in farmland plants in this study (Figure 3), and soil C:P and N:P were significantly negatively correlated with plant C:N and C:P (Figure 6), indicating that shrubland soil C:N and N:P were lower than in farmland. As a result, among these three land-use types, shrubland C:P and N:P were the lowest. N:P ratios can also indicate soil salinity [55]. This is consistent with this study's findings, which show that N:P and pH are negatively correlated (Figure 6). The shrubland's higher C:N and lower C:P and N:P indicate that plants store N in poor soils. Plants typically use storage strategies to adapt to poor habitats, most likely due to nutrient deficiency in the shrubland [56].

AN and AP in the soil are critical nutrients for crop growth. AN:AP was significantly higher in grassland than in shrubland, forestland, and farmland, which could be attributed to grassland's high AN content. AN:AP was significantly higher in forestlands than in farmlands. The variability of AN and AP in forestlands was not strong (Table 1), and both concentrations were low. This could be because forestlands are less influenced by human activities and receive less external N and P inputs from fertilizer application, whereas the application of fertilizer typically results in relatively high concentrations of AN and AP in farmland soils [57]. AP is positively correlated with TP, according to correlation analysis (Figure 6). Insoluble P dominates soil P in the watershed. Soil soluble phosphorus rises in tandem with soil phosphorus buildup [58]. The basin's agricultural soils are dominated by meadow soils and grey-cinnamon soils, and the plants of farmed meadow soils have been removed, resulting in a considerable rise in nutrient effectiveness. Within the basin, soil N:P and AN:AP showed good agreement, but N:P was higher in farmland than in shrubland,

and AN:AP was higher in shrubland than in farmland. This could be because farmland fertilizer is mostly inorganic P, which is easily converted to AP, and the AP content in the farmland is higher than that in shrubland compared with the TP content.

Soil C:N ratios tend to increase as soil depth increases. Gao et al., 2014 [14] discovered that soil C:N decreases with depth, which contradicts the current study. The increase in soil C:N indicates that humus decomposition is lower in deeper soils and humification is lower in subsoils [59]; that soil C and N losses are not proportional; and that the percentage of soil N loss is greater than that of C. As the basin is N-limited, plant N storage may occur, resulting in less N accumulation than C accumulation in the soil. Grazing is common in the basin, and causes an increase in deep soil C:N. This is because frequent trampling activities in the topsoil can significantly disrupt soil aggregates, accelerate the decomposition of soil organic matter, and increase soil susceptibility to water and wind erosion, resulting in a significant decrease in soil C storage. However, this has a weaker effect on soil N storage [53]. The soil C:P and N:P levels decreased with increasing soil depth, correlating with the findings of Hui et al., 2021 [60]. As the soil TP content was stable, the soil C and N contents decreased as the soil depth increased.

## 5. Conclusions

The findings of this study revealed that land use type has an impact on soil nutrients in alpine terrestrial ecosystems. In the surface soil, there was no discernible difference in C and N content among land types, probably due to the low nutrient content of all land types in the basin. Forestlands and shrublands contained less C and N and had a higher C:N ratio than grasslands, suggesting that thicker grass layers at higher elevations may have a greater impact on soil C and N accumulation than trees. The reduced C content and greater N and P content of farmland could be owing to tillage-induced C loss, whereas the high N and P content could be attributable to anthropogenic fertilization. The low C and N contents, C:P and N:P ratios, and high C:N ratios of shrublands indicate that shrubland organic matter decomposition is slow in the basin, N losses are significant, and shrubland soils are likely being lost, all of which should be slowed down as soon as feasible by actions such as planting. With increasing soil depth, soil C, N, and P levels, as well as C:P and N:P ratios, decrease, which could be due to the combined influence of microbes and apoplastic matter. Grazing in the basin may be responsible for the increase in soil C:N with depth. The study area was classified as a C and N limited area based on all C:N:P ratios. Management such as grazing management or moderate fertilization to promote the growth of forage and trees could help to restore the ecosystem by managing the basin pastures and forestlands.

**Author Contributions:** Conceptualization, Y.L. and Y.Y.; methodology, L.F.; writing—original, Y.L. and Y.Y.; and funding acquisition, Y.Y. and X.L. All authors have read and agreed to the published version of the manuscript.

**Funding:** This research was funded by the National Natural Science Foundation of China (Y.Y., 41871049, X.L., 41877338).

**Institutional Review Board Statement:** Not applicable.

**Informed Consent Statement:** Not applicable.

**Data Availability Statement:** The data presented in this study are available on request from the corresponding author.

**Conflicts of Interest:** The authors declare no conflict of interest.

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
