# Peer review of "Characteristics of Soil Nutrients and Their Ecological Stoichiometry in Different Land Use Types in the Nianchu River Basin"

_land, doi:10.3390/land11071001_

Round 1
Reviewer 1 Report
The research is well organised, but there are still some problems need to be solute.
1.The scientific questions need to be clarified.
2.line 123-133, the experiments need more detail description.
3.Fig 1 need to put geographic location map, and the legend should be translated into English.
4.The discussion part should be improved.
Author Response
Dear Professor
I would like to thank you very much for your valuable comments on improving the quality of this manuscript. We have carefully revised and proofread the manuscript based on your comments.
Our Responses
1) The scientific questions need to be clarified.
Answer: Thanks for your valuable comments! We have addressed all the comments carefully, and the revised portions are highlighted in green in the manuscript. We have addressed your suggestions as follows, we carefully considered and clarified the scientific question of this study(We aimed to study 1. Soil C, N, and P contents and their stoichiometry of the four land-use types in the Nianchu River basin, and 2. to assess the mechanism of the impact of land use type on soil nutrients and their stoichiometry, and 3. to summarize the implications of soil nutrient changes inthe basin for further ecological restoration) in lines 95-100.
2)line 123-133, the experiments need more detail description
Answer: Thanks for your valuable comments! We have addressed all the comments carefully, and the revised portions are highlighted in green in the manuscript. We have addressed your suggestions as follows, we add more experiment details in lines 119-158.
3) Fig 1 need to put geographic location map, and the legend should be translated into English.
Answer: Thanks for your valuable comments! We have addressed all the comments carefully, and the revised portions are highlighted in green in the manuscript. We have addressed your suggestions as follows, we add the geographic location map in line 314 and change the legend to English.
4)The discussion part should be improved.
Answer: Thanks for your valuable comments! We have addressed all the comments carefully, and the revised portions are highlighted in green in the manuscript. We have addressed your suggestions as follows, we re-examined the relevant paper to discuss the findings in lines 364 (chapter 4. Discussion).

Reviewer 2 Report
The article presents research on occurrence of nutrients in soils. Although the presented research is interesting, its description raises a number of doubts. I have proposed changes that I hope will allow the text to be improved. My recommendation is to revise the text.
General comments:
First of all, elements of scientific novelty should be clearly indicated. As it stands, the research seems to be a case study and local in scope. It seems, however, that they could also be of interest to a wider group of readers.
The nature of the environmental studies requires high standards in the description of the analytical part of the research. The method of conducting the research should not be questionable and should be clearly described. Hence the need to expand the article with detailed information on how to the analyzes have been performed. The information about the quality control should be added.
The number of significant figures (not decimal) for all data should be according with the metrological rules based on the validation of the analytical method.
Author Response
Dear Professor
I would like to thank you very much for your valuable comments on improving the quality of this manuscript. We have carefully revised and proofread the manuscript based on your comments.
Our Responses
1) First of all, elements of scientific novelty should be clearly indicated. As it stands, the research seems to be a case study and local in scope. It seems, however, that they could also be of interest to a wider group of readers.
Answer: Thanks for your valuable comments! We have addressed all the comments carefully, and the revised portions are highlighted in green in the manuscript. We have addressed your suggestions as follows, we added the importance and urgency of studying the Nianchu River basin in lines 71-83.
2)The nature of the environmental studies requires high standards in the description of the analytical part of the research. The method of conducting the research should not be questionable and should be clearly described. Hence the need to expand the article with detailed information on how to the analyzes have been performed. The information about the quality control should be added.
Answer: Thanks for your valuable comments! We have addressed all the comments carefully, and the revised portions are highlighted in green in the manuscript. We have addressed your suggestions as follows, we have added a detailed description of the method analysis in line 119 (chapter 2.2 Soil sampling and data sources).
3) The number of significant figures (not decimal) for all data should be according with the metrological rules based on the validation of the analytical method.
Answer: Thanks for your valuable comments! We have addressed all the comments carefully, and the revised portions are highlighted in green in the manuscript. We have addressed your suggestions as follows. Since organic matter and alkaline decomposition nitrogen are measured with only two decimal places, two decimal places were chosen. Checking the relevant paper (determination of valid numbers of statistics in the paper), when the measurement mean is expressed in the form of mean±standard deviation, 1/3 of the standard deviation is used to determine the number of valid digits retained, for example, 3.65 ± 0.42, whose 1/3 of the standard deviation is 0.14, and the first valid number appears after the decimal point is retained to one decimal point. The standard deviation of TP of forest land in Table 1 in this study is 0.09. Two decimal places should be retained after calculation, so two decimal places are uniformly retained in this article.

Reviewer 3 Report
The manuscript addresses an interesting topic about the differences in soil nutrient and stoichiometric characteristics of four land-use types and the relationship between soil properties and C, N, and P stoichiometry in the Nianchu River basin. Overall, the manuscript is well written, but the materials and methods, results, discussion and conclusion sections need to be improved. I have made specific comments, which are listed below.
Line 29. I suggest that the keywords are not the same as those in the title. Avoid repetition of keywords in the Title. Keywords are a tool (and another opportunity besides the title) to help indexers and search engines find relevant articles.
Line 106. For example, according to the international WRB or Soil Taxonomy classification these soils have been classified as? Please indicate the most predominant soils in this study, not just general characteristics so that potential readers can better understand.
Line 109-115. I wonder if there were any specific criteria for the selection of areas and sampling? Please provide more details about it
Line 111. In (Figure 1) the colors that represent the map are in Chinese language, please put in English language.
Line 115. More details are needed about how nutrients in plant leaves were analyzed.
Line 115-117. Report here the main types of soils analyzed
Line 134. More detail on statistical analysis needed. For example: how was the homogeneity of variance verified? The data followed a normal distribution or the dataset was transformed when not meeting the assumption of variance normality?
Line 160-162. I wonder if this result could have been more conditioned by the type of soil rather than the land-use types?
Line 163. The quality of figure 2 (a) is very low, please provide a higher resolution figure. In addition, I suggest changing the bar color of land use types (Grassland or Shrubland) is the same (Figures b, c, d, and e).
Line 163-165. Refer to the specific figure in question, e.g., here it would be Figure 2 (e). Why do you start by showing the results from the bottom up? It strikes me that your Figure 2 (a) is pH, and you have started to describe your results by figure 2 (e). Is there an explanation for this?
Line 183. I did not find reference in the text about the results of Figure 2 (a) and 2b, why?
Line 289. Why are results placed in tables and figures (Table 3 and Figure 3) at the same time? Doesn't it seem to you that this is a repetition of information?
Line 311 and 313. The Figures 5 and 6 present very low quality. Need to be improved.
Line 320. In the discussion the authors repeat part of the results. However, the discussion section should be an interpretation of your results. For example, you should compare your results with those of other studies and see if they are consistent or if not, discuss possible reasons for the difference. I suggest mentioning any inconclusive results and explaining them to the best of your ability. In addition, indicate whether your results extend the findings of previous studies.
Line 531. The conclusions are very broad. Please, remark in the conclusions the novel aspect related to this research.
Author Response
请参阅附件。

Round 2
Reviewer 2 Report
article has been improved and may be published without additionally changes
Reviewer 3 Report
The manuscript was reformulated and many of the suggestions were accepted by the authors, substantially improving fluidity. I believe that the changes made are sufficient for publication in this journal.